# Oral Health Literacy and Determinants among an Elderly Community in Portugal

**DOI:** 10.3390/ijerph21060735

**Published:** 2024-06-05

**Authors:** Helder Costa, Pedro Lopes, Maria José Correia, Patrícia Couto, Ana Margarida Silva, Joaquin Francisco López-Marcos, Nélio Veiga

**Affiliations:** 1Faculty of Medicine, University of Salamanca, 37008 Salamanca, Spain; hecosta@ucp.pt (H.C.); amsesilva@ucp.pt (A.M.S.); 2Universidade Católica Portuguesa, Faculty of Dental Medicine, 3504-505 Viseu, Portugal; paflopes@ucp.pt (P.L.); mcorreia@ucp.pt (M.J.C.); pscouto@ucp.pt (P.C.); 3Universidade Católica Portuguesa, Centre for Interdisciplinary Research in Health (CIIS), 3504-505 Viseu, Portugal; 4Surgery Department, Faculty of Medicine, University of Salamanca, 37008 Salamanca, Spain; jflmarcos@usal.es

**Keywords:** oral health literacy, elderly, oral health perception, REALD-30, GOHAI

## Abstract

High average life expectancy has caused an increase in the elderly population and with it arises the need to characterize this population regarding their health and, in particular, their oral health. The purpose of this study was to assess and characterize oral health, oral rehabilitation, oral health literacy, oral health perception and quality of life in a sample of elderly participants of a physical activity program in Portugal. An observational cross-sectional study was designed with a group of 206 individuals. All the individuals were clinically assessed, DMFT, PSR and the plaque index were registered, and a questionnaire was applied in the form of a “face-to-face” interview with questions related to the quality of life related to oral health (GOHAI index and the REALD-30 scale). Of the 206 study participants, 90.3% admit brushing their teeth daily, 6.3% practice daily flossing, and 5.8% had a dental appointment in the last 12 months. Applying the REALD-30 scale, 22.7% have a low level (score 0–14), 43.7% a moderate level (score 15–22) and 33.6% a high level (score 23–29) of oral health literacy. The GOHAI scale reveals that 37.4% have a high self-perception of their oral health. A considerable proportion of the sample studied present a moderate level of oral health literacy. Therefore, educating each person about their oral health when participating in a specific health program and developing proposals for oral health promotion activities should be widely considered as a strategy towards primary prevention. Future oral health literacy sessions should be held in order to improve oral health and quality of life among the community.

## 1. Introduction

With the advances in the field of medicine, population aging has become a trend that affects the entire world and, consequently, the health of the elderly has become an important area of concern for all countries. The World Health Organization (WHO) highlights that oral health is a key indicator of general health, quality of life and well-being. As a result, several countries have been formulating measures to improve oral health. To increase a person’s ability to speak, smile, smell, taste, touch, chew, swallow, and demonstrate feelings and emotions [1].

In the last half of the 20th century, there was an increase in the aging of the world population due to the increase in average life expectancy and the decrease in birth rates and fertility. This restructuring is due to greater access to health services, the development of healthcare, the creation of better sanitary conditions, better access and development of medicines, food and public health measures [2].

The United Nations Population Fund states that in 2012 the number of people in the world aged 60 and over was 810 million and projects that by 2050 it will be 2 billion [3].

Portugal, like the world trend, registered an increase in the aging of the population. According to the 2011 census, the population aged 0 to 14 accounted for around 15%, while the population aged 65 and over accounted for 19% [4]. Between 2015 and 2080, according to the National Statistics Institute, the resident population aged 65 or over could go from 2.1 to 2.8 million people [5].

The degradation of oral health in the elderly severely compromises their daily life, particularly in totally edentulous individuals, but also in partially edentulous individuals [6,7,8]. They tend to adopt a diet conditioned by the degree of edentulism or the adaptation and comfort of the dental prosthesis they use [9,10]. Poor oral health is linked to poor general health, and there is an increased risk of stroke in edentulous people, as well as poorer mental health [10,11].

The older adult or elderly population presents poor oral health conditions. This fact is reflected in the rate of edentulism and oral pathologies that characterize this population group. This condition must be assessed through the context in which the elderly are inserted, not worrying that the low level of oral health reflects the cumulative effect of oral pathology but also the difficulty of access to adequate dental treatment for the lifetime of these individuals [12,13]. The difficulty of the elderly population in accessing such care may be related to economic or social factors, or even due to the distribution of available resources [12].

Oral diseases are related to a state of severe morbidity with physical, psychological and social repercussions. In this way they alter the general state of health of the individual, their well-being, and their quality of life [14]. Oral health is crucial for an individual’s quality of life and is variable depending on several factors, and some behaviors are modifiable to the extent that their adoption means a reduction in oral pathology. Thus, the general acquisition of preventive behaviors increases the possibility of maintaining an adequate level of oral health, which suggests the need for training and awareness for the adoption of these behaviors and their integration at the level of social and political programs [15].

Several municipalities develop programs to adopt healthy lifestyles and promote physical activity, trying to minimize the unwanted effects of population aging [16]. The Atividade Sénior do Municipio de Viseu program (ASMV) is a municipal program developed in Portugal that appeals to healthy lifestyles with a focus on promoting daily physical activity and health literacy.

In recent years, there has been an increase in studies developed on health literacy [17]; however, the development of studies dedicated to literacy in oral health is very recent. The Rapid Estimate of Adult Literacy in Dentistry (REALD-30) is a specific tool to assess the level of literacy in oral health through the recognition of words ordered in a list with different degrees of difficulty [18,19]. This instrument is easy and quick to apply [19,20].

There are several instruments that make it possible to relate oral health to the quality of life of an individual and to perceive the existing symbiosis between these two dimensions. One of the instruments is the Geriatric Oral Health Assessment Index (GOHAI), developed specifically for the elderly population and consisting of 12 questions about the influence of oral health problems on the physical, psychosocial, and pain or discomfort dimensions of the patient [21]. The GOHAI instrument has been used over the years in diverse populations of different age groups, social strata, or with particular characteristics, such as populations with schizophrenia or cerebral palsy, and has been translated and validated into several languages such as French, Spanish and German.

The purpose of this study was to assess and characterize oral health, oral rehabilitation, oral health literacy and oral health perception in a sample of elderly participants of the ASMV program.

## 2. Materials and Methods

An observational cross-sectional study was designed through intraoral assessments and the application of a sociodemographic questionnaire in the form of a “face-to-face” interview, which includes questions related to age, gender, education, eating habits, oral hygiene habits, as well as the REALD-30 and the GOHAI scales. Data collection was accomplished between January of 2019 and December of 2019. The study design was approved by the Health Ethics Committee of the Universidade Católica Portuguesa under registration number 100.

A convenience sample of ASMV program participants were selected for the study considering the following pre-defined inclusion and exclusion criteria:Inclusion criteria: 55 years or older and less than 95 years; residents in the municipality of Viseu; prior signature of the informed consent by the patient.Exclusion criteria: people who cannot read, speak or understand Portuguese well or have cognitive problems.

Intraoral observations were performed using an observational kit (mirror and periodontal probe approved by the WHO) to identify and record the decayed, filling and missed permanent teeth index (DMFT index), the periodontal screening recorder (PSR index) and the Silness–Loe Plaque Index. The periodontal status was also recorded using a millimeter periodontal probe, which allows for a quick and effective assessment of the participant’s periodontal health status. Prior to collecting DMFT, PSR and the Silness–Loe Plaque Index data, the various examiners were properly calibrated among themselves. A total of 10 individuals were selected and were individually evaluated by each of the examiners and the results obtained were duly recorded. Subsequently, the results were compared and a Kappa value = 0.93 was obtained.

To assess the quality of life related to oral health, the GOHAI index was applied. The GOHAI, which consists of 12 questions about the influence of oral health problems on the patient’s physical, psychosocial, and pain or discomfort dimensions, was chosen because it has been developed specifically for the elderly population. Physical function is represented by the pattern of chewing, speaking and swallowing, psychosocial function, represented by concern for oral health, satisfaction or dissatisfaction with appearance, self-awareness about their oral health and avoidance of social contact due to problems and pain or discomfort, represented using medication to relieve pain or discomfort. Each participant was asked each of the 12 questions that make up the GOHAI index with the possibility of answering always, sometimes or never. Each answer was assigned a value: 1 = always, 2 = sometimes and 3 = never, with the sum of all the questions being made at the end. Thus, three intervals of self-perception of oral health were considered according to the total score obtained: high (34–36), moderate (30–33) and low (<30) self-perception. The lower the score, the lower the level of positive perception of their oral health, therefore, the worse the level of quality of life related to the oral health of the elderly.

The assessment of the level of oral health literacy of the sample was carried out using the REALD-30 scale, which consists of a list of 30 words that the individual must read aloud to the interviewer. The instrument has an evaluation from 0 to 30, assigning 1 point for each correctly pronounced word. Previously, the individual is instructed to say “I do not know” when they cannot read one of the words. At the end, the sum of all the points is made, where 0 represents low literacy and 30 represents higher literacy.

All the questionnaires were applied and completed by the interviewer based on the study participant’s response and recorded directly in the database using one of the computing devices (tablet, mobile phone or computer). The purpose of the questionnaires was previously explained, and the informed consent form was signed.

The statistical analysis of the data was carried out with the statistical software IBM-SPSS^®^ 24.0 (IBM, Chicago, IL, USA). Measures of central tendency (mean and standard deviation) and prevalence, expressed as a percentage, were calculated. The respective hypothesis tests necessary for the realization of the inferential statistics were also carried out, using a significance level of 5%.

## 3. Results

The final study sample was composed of 206 individuals, of whom 69.4% were women (n = 143) and 30.6% were men (n = 63); a total of 21.8% participants (n = 45) were under 65 years of age and 78.2% participants (n = 161) are 65 years or older with a mean age of the participants of 70.0 ± 7.16.

Regarding the scholarship level, 39.8% participants (n = 82) completed the 9th grade, 3.9% (n = 8) the 12th grade and 3.4% (n = 7) completed a higher education level.

Table 1 demonstrates the assessment of oral health behaviors of the study sample and the levels of oral health literacy obtained through the application of the REALD-30 scale. Of the 206 participants in the study, 90.3% admit brushing their teeth daily. Of these, 31.7% brush once a day, 46.8% twice a day and only 21.5% brush three or more times a day. As for completing daily brushing with flossing, only 6.3% of participants admitted to doing so. When asked when was the last time they had a dental appointment, 55.8% of the sample answered that it was less than 12 months ago and 82.5% say they needed a dental appointment.

Regarding the levels of oral health literacy, by applying the REALD-30 scale to our sample, we can say that 22.7% have a low level (score 0–14), 43.7% a moderate level (score 15–22) and 33.6% a high level (score 23–29).

Table 2 presents the characterization of the sample according to the DMFT index. An average DMFT of 10.38 ± 8.55 was recorded in the sample. Within this DMFT value, we have an average of 4.52 filled teeth, 11.96 missing and 2.48 decayed. The highest score corresponded to the missing teeth component 11.96 ± 8.56.

Table 3 describes the characterization of the Silness–Loe plaque index. It can be observed that a prevalence in the sample of 32% was recorded with “value 2”, which refers to a moderate layer of plaque near the gingival margin and free interdental spaces visible to the naked eye.

Table 4 describes the characterization of the sample according to the Periodontal Screening and Recording Index (PSR index). It can be observed that approximately 30% of the study sample have periodontal health, and that approximately 62% (code 3 and 4) have periodontal problems, with pockets greater than 3.5 mm that require periodontal therapy.

Table 5 presents the clinical data from the sample, where it is important to point out that 22.8% of the sample reported having a toothache at the time, 38.8% assumed that they had a sensation of dry mouth (xerostomia) and 43.9% stated that they used dental prostheses.

Regarding the diagnosis of other oral pathologies, it is important to point out that in the present study the following prevalence values were detected (Table 6).

Table 7 describes the self-perception of oral health through the GOHAI index. The lower the score, the lower the level of positive perception of their oral health, therefore, the worse the oral health of the elderly. Of the participants who responded, 37.4% have a high self-perception of their oral health, 32% moderate, and 30.6% have a low self-perception of their oral health.

In terms of inferential statistics, we can verify that there are significant statistical differences:

Gender was associated with daily toothbrushing (female = 93.7% vs. male = 82.5%, *p* = 0.013) and the use of dental prothesis (female = 39.4% vs. male = 54.1%, *p* = 0.039).

Age was associated with the frequency of a dental appointment in the last 12 months (under 65 = 73.3% vs. 65 or more = 50.9%, *p* = 0.024) and the use of a dental prothesis (under 65 = 57.5% vs. 65 or more = 40.5%, *p* = 0.04).

Table 8 describes the relationship between the level of oral health literacy and the use of dental prostheses. We also observed that those with the highest level of oral health literacy (3rd percentile) were also those who used dental prostheses the most (46%, *p* = 0.84).

Table 9 presents the inferential statistics of the GOHAI score and DMFT score, and we can verify that of the participants with DMFT equal to or less than 8, 34.3% have a high self-perception of their oral health and of those with DMFT greater than 8, 40.6% have the same level of self-perception.

We can verify that of the participants who have a high self-perception of their oral health, only 3.9% report having toothache, while among the participants with a low self-perception of their oral health, 42, 9% report having a toothache (*p* < 0.001).

When analyzing the association between the periodontal status and other variables, we verified that there is a significant statistical difference between the diagnosis of severe periodontitis and toothache episode (*p* = 0.003) and moderate periodontitis and the GOHAI scores (*p* = 0.034).

## 4. Discussion

In the present study, 90.3% of the individuals admit to brushing their teeth daily, 46.8% admit brushing them twice a day and only 6.3% perform interdental hygiene. The results obtained are lower than those found in a study for the Portuguese population, in which 97.3% admit brushing their teeth daily, 73.4% do so twice a day and 23.8% state that they perform interdental hygiene [22].

It is in the female gender that there is a greater habit of daily brushing (93.7% vs. 82.5%, *p* = 0.013) and the same trend is verified in interdental hygiene (7.7% vs. 3.2%, *p* = 0.074). These results are similar to those obtained in the study by Melo P et al. of the Portuguese population [22], in which women have a higher prevalence of daily brushing than men (98.1% vs. 96.3%, *p* = 0.069) as well as performing interdental hygiene (10.7% vs. 3.4%, *p* = 0.024).

In our study, we observed that 55.8% of the sample had their last dental appointment less than 12 months ago, while in the general Portuguese population, 47.4% had not visited a dentist in the last 12 months. The authors of the study pointed out as the most probable causes the economic factor and the lack of integrated attention to oral health in the national health services [22].

Many studies in the literature refer to the lack of perception of the need for dental treatments as the main reason why people do not visit the dentist regularly. The non-perception arises mainly due to the lack of pain, the total or partial absence of teeth or simply because they find the frequent dental check-up appointment unnecessary [23]. In the present study, we did not verify this, because nearly 82.5% of the individuals stated that they felt the need for a dental appointment in the moment of the study.

Regarding the oral hygiene index observed in the sample, it was found that 19% did not present dental plaque in the intraoral observation (Value 0), 30.6% had “Value 1”, 32% “Value 2” and 18.4% “Value 3”. A study carried out on a sample of Portuguese individuals also obtained similar results with 16.4% of the sample without plaque (Value 0), 20.5% “Value 1”, 58.2% “Value 2” and 4.9% “Value 3” [24]. In both studies, “Value 2”, which corresponds to the presence of visible plaque, was the most prevalent. There is an association between the presence of dental plaque and periodontal disease [25,26]. In the present study, we observed that 58.3% had periodontitis (22.8% mild, 21.4% moderate, 14.1% severe) and 12, 1% gingivitis. The prevalence of periodontitis is relatively higher than the results obtained by Revas M et al. [27], who recorded in his study that 48.6% of the sample had periodontitis and 26.9% gingivitis.

Regarding the DMFT index, we found an average value of 10.38 in our sample. These values are in line with those obtained in the study related to the Portuguese population, where a mean value of the DMFT index was recorded for the age group 65–74 years of 15.11 [28].

The value obtained in our study is substantially lower than the value found in the systematic review carried out by Chan et al. [29], where there was a global average value of the DMFT index of 21.9%. This difference between our results and the study developed by Chan et al. is probably due to the fact that oral healthcare and oral hygiene habits might be slightly different in Portugal and, in comparison, with other countries.

In Portugal, the lack of oral health literacy is a serious public health issue. We still find in our population considerable levels of lacking health literacy, with consequences for health in general and oral health in particular [30]. Therefore, the application of the appropriate scales to assess oral health literacy is important to establish adequate health education strategies in the community. It is widely accepted that low health literacy is related to problems with the use of preventive services, late diagnosis of medical conditions, poor adherence to medical instructions, poor self-management skills, increased mortality risks, poor health outcomes and higher health care costs [17]. The results we obtained between the level of literacy and the use of dental prosthesis may suggest that those who have a higher level of literacy in oral health seek and adhere better to oral health treatments.

In the results of the GOHAI scale, we can verify that 37.4% have a high self-perception of their oral health, 32% a moderate, and 30.6% have a low self-perception of their oral health. The values are slightly different from those obtained in the study carried out by Carvalho et al., where 59.9% of the participants presented a high self-perception of their oral health, 27.8% a moderate, and 12.3% a low self-perception [31]. Also, in a study carried out by Osta et al., 56% presented a high self-perception of their oral health, 47.8% a moderate, and 34.5% a low self-perception [32]. In association, the results of this study reveal the need for developing more and efficient oral health promotion strategies and community interventions, mainly among the elderly and the most vulnerable members of the population.

As limitations of the study, we can consider the difficulty of obtaining a second group with different characteristics, not belonging to the ASMV program, which would allow us to make comparisons between different groups and evaluate the differences or similarities in the results. Another of the participants’ de facto limitations may be the overestimation of their real behaviors and oral hygiene habits, which may have introduced biases in data collection, with no way of determining or measuring this influence.

## 5. Conclusions

A considerable proportion of the sample studied present a moderate level of oral health literacy. Therefore, the development of health programs for the community, and focused on specific risk groups, is essential because they enable the possibility of diagnosing oral health behaviors, oral status, and understanding the needs of the elderly population. It also permits more frequent contact with health professionals. Educating each person about their oral health when participating in a specific health program and developing proposals for oral health promotion activities should be widely considered the main goals in primary prevention, mainly among the elderly over 65 years of age, where we can verify worse oral health behaviors and quality of life. Future oral health literacy sessions should be held in order to improve oral health and quality of life among the community.

## Figures and Tables

**Table 1 ijerph-21-00735-t001:** Oral health behaviors and oral health literacy as assessed by REALD-30 of the sample.

Oral Health Behaviors	n	%
Daily brushing		
Yes	186	90.3
No	20	9.7
Daily brushing: number of times per day		
1 time/day	59	31.7
2 times/day	87	46.8
3 or more times/day	40	21.5
Daily interdental hygiene		
Yes	13	6.3
No	193	93.7
Last dental appointment		
The last appointment was less than 12 months ago	115	55.8
The last appointment was more than 1 year ago and less than 2 years ago	39	18.9
The last appointment was more than 5 years ago	52	25.3
Need for a dental appointment		
Yes	170	82.5
No	36	17.5
REALD-30 classification		
1st percentile (0–14)—Low	47	22.7
2nd percentile (15–22)—Moderate	90	43.7
3rd percentile (23–30)—High	69	33.6

**Table 2 ijerph-21-00735-t002:** Characterization of the DMFT index.

	N	Minimum	Maximum	Average	Standard Deviation
DMFT	206	0	28	10.38	8551
Filled	126	1	19	4.52	3372
Missing	105	1	28	11.96	8564
Decayed	126	1	7	2.48	1643

**Table 3 ijerph-21-00735-t003:** Characterization of the Silness–Loe plaque index.

Silness–Loe Plaque Index	n	%
0	39	19.0
1	63	30.6
2	66	32.0
3	38	18.4

**Table 4 ijerph-21-00735-t004:** Characterization of the PSR index.

PSR Index Codes	n	%
0—Periodontal health	62	30.1
1—Bleeding on probing	19	9.2
2—Calculation detected	82	39.8
3—Periodontal pocket 3.5–5.5 mm	115	55.8
4—Periodontal pocket 6 mm or larger	13	6.3

**Table 5 ijerph-21-00735-t005:** Clinical data.

Clinical Data of the Sample	n	%
Toothache		
Yes	47	22.8
No	159	77.2
Sensation of dry mouth (xerostomy)		
Yes	80	38.8
No	126	61.2
Taste disturbance		
Yes	41	19.9
No	165	80.1
Use of dental prosthesis		
Yes	87	43.9
No	111	56.1

**Table 6 ijerph-21-00735-t006:** Prevalence of oral diseases.

Oral Pathology	%
Gingivitis	12.1%
Mucositis	2.9%
Periodontitis (initial, mild)	22.8%
Periodontitis (moderate)	21.4%
Periodontitis (severe)	14.1%

**Table 7 ijerph-21-00735-t007:** GOHAI Scores.

GOHAI	No	%
High self-perception (34–36)	77	37.4
Moderate self-perception (30–33)	66	32.0
Low self-perception (<30)	63	30.6
I do not answer	109	52.9

**Table 8 ijerph-21-00735-t008:** Inferential statistics REALD score and the use of dental prosthesis.

REALD-30	Percentile 1 (0–14) Low		Percentile 2 (15–22) Moderate		Percentile 3 (23–30) High	
	n	%	n	%	n	%
Use of dental prosthesis						
Yes	19	42.2	36	41.4	29	46
No	26	57.8	51	58.6	34	54

**Table 9 ijerph-21-00735-t009:** Inferential statistics GOHAI score and DMFT.

	DMFT Less than or Equal to 8		DMFT Greater than 8	
	n	%	n	%
GOHAI				
High self-perception (34–36)	36	34.3	41	40.6
Moderate self-perception (30–33)	33	31.4	33	32.7
Low self-perception (<30)	36	34.3	27	26.7

## Data Availability

The original contributions presented in the study are included in the article, further inquiries can be directed to the corresponding author.

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
