# Peer review of "Oral Health Literacy and Determinants among an Elderly Community in Portugal"

_ijerph, 2024, doi:10.3390/ijerph21060735_

Round 1
Reviewer 1 Report
Comments and Suggestions for Authors
Thanks for the opportunity to read and suggest some improvements to the authors of the article: Oral health literacy and determinants among an elderly community.
In the introduction they should address the use and experiences of using the Geriatric Oral Health Assessment Index (GOHAI-SP) in other populations.
The objective should be more than assess and characterize oral health, oral rehabilitation-91 tion, oral health literacy and oral health perception in a sample of elderly participants of the ASMV program.
Since it uses different instruments to assess the oral health of people in the ASMV program, it must correlate the results of these different instruments so that it is considered research and not just a report.
I suggest removing secondary objectives.
InMethods:
You should describe your sampling method, for example: A convenience sample of participants.
The use of indices such as DMFT, PSR and the Silness-Loe Plaque Index must go through calibration of the subjects who recorded the lesions or plaque. The criteria should also briefly describe the characteristics of the conditions under which the data was recorded.
Results:
In Table 2. Characterization of the PSR index.
Calculation???
The authors must include DMFT, PSR and dental plaque index values in a table.
Analyzes comparing the values of GOHAI Scores, REALD-30 with DMFT PSR and are missing. As well as develop a predictive model or another inferential analysis.
Author Response
Reviewer 1:
In the introduction they should address the use and experiences of using the Geriatric Oral Health Assessment Index (GOHAI-SP) in other populations.
We would like to thank the reviewer for the comments and the clear and exhaustive suggestions to improve the quality of the manuscript. We added this information on line 90 to 93.
The objective should be more than assess and characterize oral health, oral rehabilitation, oral health literacy and oral health perception in a sample of elderly participants of the ASMV program.
We thank the reviewer for your comment. In this study, we only aimed to analyze these parameters. However, we are preparing a new, more comprehensive study, in which we will analyze more variables in order to establish future strategies towards oral health literacy.
Since it uses different instruments to assess the oral health of people in the ASMV program, it must correlate the results of these different instruments so that it is considered research and not just a report.
In results, we introduce a new table, number 6, which includes the correlation between the results of the REALD 30 instrument regarding the level of oral health literacy and the use of dental prostheses in our sample (line 205 to 211). Also in the discussion, line 269 to 272 we include this correlation.
I suggest removing secondary objectives.
We thank the reviewer for your comment. This information was removed from the manuscript in introduction.
In Methods: You should describe your sampling method, for example: A convenience sample of participants.
As suggested, the description of the convenience sample was included in line 104.
The use of indices such as DMFT, PSR and the Silness-Loe Plaque Index must go through calibration of the subjects who recorded the lesions or plaque. The criteria should also briefly describe the characteristics of the conditions under which the data was recorded.
As suggested, we include the explanation of calibration process for examiners in lines 114 to 118.
Results
In Table 2. Characterization of the PSR index. Calculation? The authors must include DMFT, PSR and dental plaque index values in a table.
As suggested for better reading of the results, a table was placed for DMFT values (table 2) and plaque index (table 3).
Analyzes comparing the values of GOHAI Scores, REALD-30 with DMFT and PSR are missing. As well as develop a predictive model or another inferential analysis.
Following the suggestion, table 9 was added with the inferential statistics of the GOHAI score and the DMFT Score.
Reviewer 2 Report
Comments and Suggestions for Authors
1. Brief Summary and General Comments
Dear authors,
Thank you for providing me with the opportunity to review your manuscript. I would like to congratulate you on your interesting cross-sectional study about the oral health literacy and determinants among an elderly community in Portugal. In my opinion, the topic would be of relevance to the International Journal of Environmental Research and Public Health because it addresses oral health, rehabilitation status, oral health literacy, oral health perception, and the relation to the quality of life of elderly people. I hope my feedback will be helpful for the revision of the manuscript. Please consider the following comments:
2. Title and Abstract
- Title: After having read the title of the manuscript on its own it left me with the question about which methodology had been used and where the study had been conducted. Therefore, I would recommend adding the study design and the country to the title for clarification to the reader.
- Abstract: Overall, the abstract gives a clear picture of the study, however, I would suggest adding the place where and the time when the study was conducted, and the indices used (e.g., line 17: “All the individuals were clinically assessed…”) for clarification to the reader.
3. Introduction
- The introduction is informative and well structured. Please clarify that the “Atividade Sénior do Municipio de Viseu” is a Portuguese program if I am understanding it right (line 74-78).
- Regarding the REALD-30 scale and the GOHAI index, are there any research results on elderly available that could be added to the introduction?
- Given the quantitative study design, I would suggest adding the hypotheses to the aim of the study.
4. Methods
- The methods section gives the reader a quite good idea about the study though several information could be added to allow for a replicability by other researchers if desired.
- Inclusion/exclusion criteria: Were there any restrictions regarding the general health status of the participants applied, or could everyone irrespective of the general health status participate in the ASMV program? How many participants did the ASMV program have in total? How was the recruitment performed? Was the number of participants based on a sample size calculation?
- When was the study conducted? Please add the study period.
- Concerning the clinical examination, I would recommend adding information about the calibration training of the examiners. How many examiners participated in the study?
- Please add all indices used to the methods section (e.g., the PSR and OHI are only mentioned in the results section).
- Please add information about the validity of the questionnaire.
- Statistical analysis: Please complete the information about the software (manufacturer, city, (state), country) and add which statistical tests were used.
5. Results
- The results are described clearly and relate back to the objectives. For more transparency to the reader, I would recommend adding the dropout rate and explaining why number of participants presented in certain categories deviates from n = 206 (e.g., scholarship level, daily brushing, use of dental prothesis).
6. Discussion and Conclusions
- The discussion section links the findings to other studies conducted within the Portuguese population. I would have liked to read more about whether the different cohorts allow for a comparability (e.g., similarities/disparities between the cohorts) and potential reasons for differences in the results, which could have also been discussed in a broader context by adding results from studies conducted in other countries.
- The limitations of this study should be addressed for clarification to the reader (e.g., representativeness of the sample, generalizability of results).
7. References
- Please check the references to make sure that they are aligned with the submission guidelines. For example, articles and websites should be cited as follows:
Journal Articles: Author 1, A.B.; Author 2, C.D. Title of the article. Abbreviated Journal Name Year, Volume, page range
Example: 13. Gomes M, et al. Clinical Predictors of Quality of Life Related to Oral Health in Idols with Diabetics. Reference Nursing Journal 320 . 2015;IV (7):81–9.
Websites: Title of Site. Available online: URL (accessed on Day Month Year)
Example: 2. World Health Organization. World report on Aging And Health [Internet]. 2015 [cited 2019 Apr 2]. Available from: 298 https://apps.who.int/iris/bitstream/handle/10665/186463/9789240694811_eng.pdf?sequence=1
8. English language
- Overall, the manuscript could benefit from some minor editing of English language (grammar).
Based on these comments, I would recommend revising and re-reviewing the manuscript. Thank you in advance for considering the comments during revision. Good luck to the authors and kind regards!
Comments on the Quality of English LanguageOverall, the manuscript could benefit from some minor editing of English language (grammar).
Author Response
Reviewer 2
- Brief Summary and General Comments
Dear authors,
Thank you for providing me with the opportunity to review your manuscript. I would like to congratulate you on your interesting cross-sectional study about the oral health literacy and determinants among an elderly community in Portugal. In my opinion, the topic would be of relevance to the International Journal of Environmental Research and Public Health because it addresses oral health, rehabilitation status, oral health literacy, oral health perception, and the relation to the quality of life of elderly people. I hope my feedback will be helpful for the revision of the manuscript. Please consider the following comments:
- Title and Abstract
Title: After having read the title of the manuscript on its own it left me with the question about which methodology had been used and where the study had been conducted. Therefore, I would recommend adding the study design and the country to the title for clarification to the reader.
Abstract: Overall, the abstract gives a clear picture of the study, however, I would suggest adding the place where and the time when the study was conducted, and the indices used (e.g., line 17: “All the individuals were clinically assessed…”) for clarification to the reader.
As suggested, the location and the main indices recorded were included in the abstract on lines 16, 17 and 18.
- Introduction
The introduction is informative and well structured. Please clarify that the “Atividade Sénior do Municipio de Viseu” is a Portuguese program if I am understanding it right (line 74-78).
Thanks for the comment. Yes, the senior activity program is a Portuguese program we clarify the information on line 76.
Regarding the REALD-30 scale and the GOHAI index, are there any research results on elderly available that could be added to the introduction?
We thank the reviewer for your comment. We presented some studies in the Discussion section.
Given the quantitative study design, I would suggest adding the hypotheses to the aim of the study.
We thank the reviewer for your comment. We decided not to define hypotheses but only orientate the results and conclusion in order to answer the objectives of the research.
- Methods
The methods section gives the reader a quite good idea about the study though several information could be added to allow for a replicability by other researchers if desired.
Inclusion/exclusion criteria: Were there any restrictions regarding the general health status of the participants applied, or could everyone irrespective of the general health status participate in the ASMV program? How many participants did the ASMV program have in total? How was the recruitment performed? Was the number of participants based on a sample size calculation?
We thank the reviewer for your comment. In fact, there were no restrictions on participation in the ASMV program with regard to general health status. A convenience sample was considered for the study, this information was added to the manuscript on line 104.
When was the study conducted? Please add the study period.
We thank the reviewer for your comment. We have added the information in the manuscript on line 101 and 102.
Concerning the clinical examination, I would recommend adding information about the calibration training of the examiners. How many examiners participated in the study?
This information was added to the manuscript on line 114 to 118.
Please add all indices used to the methods section (e.g., the PSR and OHI are only mentioned in the results section).
This information was added to the manuscript in the methods section.
Please add information about the validity of the questionnaire.
We thank the reviewer for your comment. The questionnaire was validated for the portuguese population and the validation process is described in a paper also published by us (Costa H, Amaral O, Duarte J, Correia MJ, Veiga N, López-Marcos JF. Validity and reliability of the Portuguese version of the REALD-29 PT. BMC Oral Health. 2022; 22:262. https://doi.org/10.1186/s12903-022-02289-w.
Statistical analysis: Please complete the information about the software (manufacturer, city, (state), country) and add which statistical tests were used.
We thank the reviewer for your comment. The information was added on the manuscript on lines 145 and 146.
- Results
The results are described clearly and relate back to the objectives. For more transparency to the reader, I would recommend adding the dropout rate and explaining why number of participants presented in certain categories deviates from n = 206 (e.g., scholarship level, daily brushing, use of dental prothesis).
We thank the reviewer for your comment. We do not have a dropout rate specifically because we applied a convenience sample and the individuals would come to the research group and participate if they wanted to, voluntarily. Some deviation of categories is do to the fact of some participants not wanting to answer those specific questions.
- Discussion and Conclusions
The discussion section links the findings to other studies conducted within the Portuguese population. I would have liked to read more about whether the different cohorts allow for a comparability (e.g., similarities/disparities between the cohorts) and potential reasons for differences in the results, which could have also been discussed in a broader context by adding results from studies conducted in other countries.
We thank the reviewer for your comment. We have in the Discussion comparison with some studies similar to ours. We found it difficult to compare more studies because of the difference of methodologies found in other studies.
The limitations of this study should be addressed for clarification to the reader (e.g., representativeness of the sample, generalizability of results).
This information was added to the manuscript on line 296 at 301.
- References
Please check the references to make sure that they are aligned with the submission guidelines. For example, articles and websites should be cited as follows:
Journal Articles: Author 1, A.B.; Author 2, C.D. Title of the article. Abbreviated Journal Name Year, Volume, page range
Example: 13. Gomes M, et al. Clinical Predictors of Quality of Life Related to Oral Health in Idols with Diabetics. Reference Nursing Journal 320 . 2015;IV (7):81–9.
Websites: Title of Site. Available online: URL (accessed on Day Month Year)
Example: 2. World Health Organization. World report on Aging And Health [Internet]. 2015 [cited 2019 Apr 2]. Available from: 298 https://apps.who.int/iris/bitstream/handle/10665/186463/9789240694811_eng.pdf?sequence=1
We thank the reviewer for your comment. The correction was made on the manuscript.